

# If a Rainfall–Runoff Model was a Hydrologist

John Ewen, Greg O'Donnell

School of Engineering, Newcastle University, Newcastle upon Tyne, NE1 7RU, UK

*Correspondence to*: Greg O'Donnell (g.m.odonnell@newcastle.ac.uk)

**Abstract.** Personification can be annoying, but also instructive. If a Rainfall-Runoff (RR) model was a hydrologist it could be called the Modelled Hydrologist, MH. Ideally, an MH used when tackling real–world problems such as flooding and climate change would be acquainted with hydrologic laws at the catchment scale and with a diverse panel of desk and field hydrologists who have between them thousands of years of experience. In practice, though, the MHs for RR models are largely ignorant of hydrology. Some of this ignorance is real (e.g. the hydrologic laws are unknown). The rest is selective ignorance, as is
practised throughout science whenever there is a need for complex system analysis, parsimony, or similar. It is a form of designed neglect. In lumped RR modelling, the classic MH is that used in Jakeman and Hornberger's experiment on their question "How much complexity is warranted in a rainfall–runoff model?". Based on what that MH "knows" it is a statistician dilettante–hydrologist. When studying difficult and confusing problems (conundrums) like RR modelling it is helpful to have simple concrete examples to use as benchmarks and when generating hypotheses. Here, an MH for lumped modelling is built
which is a layman with an interest in the weather and river flows (e.g. a river fisherman). The MH is created in a novel experiment in which statements of knowledge in everyday English are transformed systematically into a novel parameterless RR model. For a set of 38 UK catchments, the relative importance is measured as 1 and 6, respectively, for the layman's knowledge about seasonality and wetness, and 2 for the knowledge that runoff records are always unpredictable to some degree. Hydrologic laws are discussed and hydrologic similarly in time and place is explored.

## 1 Introduction

Rainfall–Runoff (RR) modelling is sometimes used when making life and death decisions about drought, flood, land use, soil erosion, pollution, and climate change. Over the last few decades there have been many notable contributions to discussions on the difficulties and dangers inherent in RR modelling. These include: (1) Dooge's (1986, p.50S) observation that hydrologic laws at the catchment scale should amount to more than "mere data fitting"; (2) The comments from Klemeŝ (1986a) on
"dilettantism in hydrology"; (3) Kirchner's (2006) memorable comment that "models ... may succeed as mathematical marionettes, dancing to match the calibration data even if their underlying premises are unrealistic"; (4) Beven's writings on the notion of "equifinality", defined in his textbook (Beven, 2012) as "the concept that there may be many models of a catchment that are acceptably consistent with the observations available", and his writings on parameter uncertainty (e.g.





Beven, 1993); and (5) Jakeman and Hornberger's (1993) experiment on their question "How much complexity is warranted
in a rainfall–runoff model?".

There is huge scope for drawing *hydrologic conclusions* on the basis of RR modelling:  e.g. (1) the values in a given simulated
runoff time series for catchment X are good as estimates to fill the gaps in the runoff record for catchment X; (2) the effect on
peak runoff resulting from planting forest W in catchment X is Y; (3) it can be proved that there was snowmelt on 8[th] January
1962 in catchment X even though the collection of runoff records did not start until 1970; and (4) for catchment X, 14[th] June
2002 is hydrologically similar to 28[th] May 1985.  In engineering hydrology, one role for such conclusions is to support decisions.
As noted by Baker (2017): "engineering hydrology incorporates the best available theory (science–as–knowledge) into models
in order to achieve accurate representation (simulation) of the system of interest for some problem of control to be solved
within limitations of time and available resources".  The management of drought, flood, land use, etc are clearly problems of
control which must be solved within limitations of time and available resources.  There are substantial deficits in what is
labelled above as science–as–knowledge.  In his famous lectures on natural laws, Richard Feynman (1967) concluded that the
simplicity and beauty of physical laws is linked directly to the fact that they are mathematical.  Catchment RR science, however,
is still at the stage of relying on common sense, statistical methods, and small–scale (essentially point–scale) physics.  Note
that problem 6 in the current list of 23 unsolved problems in hydrology is as follows:  "What are the hydrologic laws at the
catchment scale and how do they change with scale?" (Blöschl et al., 2019).  For the purpose here it is convenient to use the
general term *hydrologic knowledge* to denote hydrologic things which are (or are treated as if they are) widely thought true or
valid by hydrologists.  It can therefore be said (and is depicted in Panel A in Fig. 1) that decisions depend on hydrologic
conclusions which in turn depend, via RR modelling, on hydrologic knowledge.  Panel A does not show feedback loops (e.g.
a loop back from performance if there was parameter calibration).  Panel B in Fig. 1 will be discussed later.



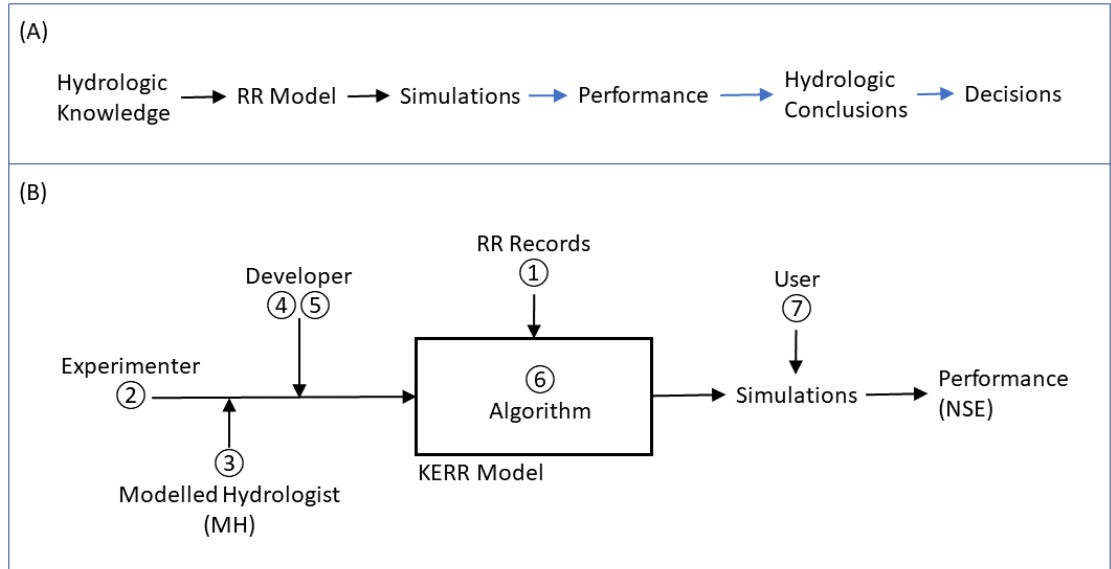

**Figure 1 Hydrologic knowledge flow in the absence of feedback (Panel A) and within the knowledge experiment, showing the table**
**numbers (Panel B)**

There are real–world risks associated with the flow of hydrologic knowledge. The obvious risk is that something to the right of "hydrologic knowledge" in Panel A in Fig. 1 might not be justified by the hydrologic knowledge employed. The inverse of this is also relevant: i.e. false hydrologic knowledge might be "discovered" which is not justified by the simulations it helps

create (false conclusions and poor decisions can result from false discoveries). These risks cannot be addressed using traditional split–sample validation testing (a fact well understood by Klemeŝ, 1986b). The following are two, relevant, appropriate, concrete examples of ways that have been used to deal with these risks: (1) the "blind validation method" developed by Ewen and Parkin (1996) forces the modellers to focus on knowledge rather than data fitting when predicting the effects of changes in land use and climate (this method was developed in work on the use of RR modelling in post–closure

radiological safety assessments of deep geological repositories for solid radioactive wastes); and (2) Jakeman and Hornberger (J&H; 1993) used sophisticated statistical techniques to help avoid making false discoveries about the complexity warranted when simulating runoff hydrographs.

The work here is exploratory scientific work driven by curiosity about: (1) the role played by hydrologic knowledge in RR

modelling; and (2) how hydrologic knowledge flows within RR modelling (e.g. if it is lost or corrupted, exactly where does



that happen, and how). Experience shows that understanding the flow of hydrologic knowledge is a difficult and confusing problem (a conundrum). Scientific hydrology, like all science, is an activity and attitude rather than a method or a set of knowledge or a set of rules. When there are significant deficits, difficulties or dangers, one of the most powerful tools for exploratory work in science is the construction of simple, complete, concrete examples. Such examples can serve as references,
benchmarks, exemplars, as the basis for generating hypotheses, or simply as supports (e.g. support of the type given in Pfister and Kirchner, 2017, where their commentary on the nature and value of hypothesis testing in hydrology gained significantly in strength and clarity simply by citing four concrete examples). The notion of using examples is entirely consistent with something which for convenience will be called *selective ignorance*. Science is always practised with some degree of selective ignorance. This is not wilful neglect in support of the unsupportable, but rather the designed neglect implicit in what Mitchell
(2009, p. 38) described as *idea models*: "...models that are simple enough to study via mathematics or computers but that nonetheless capture fundamental properties of natural complex systems".

A simple, complete, concrete example is created transparently here, albeit for a simple RR model. A slightly disruptive, idiosyncratic approach is adopted: after all, there would be little point in driving towards a long–standing conundrum along
ruts in the road. The difficulties and dangers mentioned earlier arise because the flow in Panel A in Fig. 1 is not well behaved; one problem is the feedback mentioned earlier, and another is "cherry–picking" (Pfister and Kirchner, 2017) where researches and engineers see and use what they want to see and use. In the concrete example, the aim is that the knowledge flow is simple and well behaved: the simulations are derived systematically from the hydrologic knowledge in a way which is as close as practical to deduction. It is a little grand to claim that the work falls within the general scope of what has been called the
"fourth paradigm for hydrology" (data–intensive studies of scaling and similarity, Peters–Lidard et al., 2017), but it does exploit similarity within RR records and exploit scale–free similarity between the RR records for different catchments. Any type of RR model could probably be used in work on the flow of hydrologic knowledge. The ideal modelling, though, would not rely on data fitting and would have a direct, explicit representation of hydrologic knowledge. No suitable parameterless model exists. Some models do incorporate knowledge explicitly, including fuzzy models (e.g. Seibert and McDonnell, 2002).
For example, the fuzzy–autoregressive model for daily river flows developed by Greco (2012) has linguistic statements of simple hydrologic knowledge, one of which is: "If inflow rate is low, then surface runoff is low". As a means to an end here (i.e. to meet the requirements for working with the flow of knowledge, and not just for the sake of developing yet more, or better, or different RR models), simple parameterless RR models are developed from scratch based directly on statements of hydrologic knowledge.


## 1.1 Aim and Layout

People who find personification annoying will be disappointed to discover that the idiosyncratic approach adopted here involves thinking of an RR model as a hydrologist (called the *Modelled Hydrologist*, MH) who works in concert with a real



hydrologist (the *model user*) who applies and runs the model. Ideally, an MH would be acquainted with hydrologic laws at
the catchment scale and with a diverse panel of desk and field hydrologists who have between them thousands of years of
experience. In creating the concrete example, care is taken in designing and studying the selective ignorance for a novel MH.
The nearest similar work is J&H's experiment, in that the work here can be interpreted as being about the hydrologic knowledge
warranted in RR modelling. The case for saying that this work addresses the problem of understanding the flow of hydrologic
knowledge is that: (1) hydrologic knowledge is made explicit and is measured; (2) considerable efforts are made to minimise
the loss or corruption of hydrologic knowledge in the modelling; and (3) efforts are made to understand the information in RR
records in terms of their direct usefulness to hydrologists, specifically the need to draw valid hydrologic conclusions from
using RR models. Some definitions are given in Sect. 2 and the method is outlined in Sect. 3. There is some detailed theory
in Sect. 4 about the numerical method used. The data set is introduced in Sect. 5 and an experiment on knowledge (i.e. a
knowledge experiment) is designed in Sect. 6 and run in Sect 7. The work is summarised in Sect. 8, and discussed in Sect. 9.

**2 Definitions**

Given that the aim is to keep everything as simple as practical, it is appropriate to work with the *minimum data set* (i.e. the set
comprising the catchment–average rainfall and the runoff at the catchment outfall, plus the catchment area if the records do
not have the same physical units). *Transposability in time* relates to using one part of a runoff record to predict another part
(Klemeŝ, 1986b). *Transposability in place* relates to using the RR record from one catchment when simulating the runoff for
another catchment. Note that the problem of prediction for ungauged catchments can be approached as a problem in
transposability in place (Hrachowitz et al., 2013).

There must be *knowledge hygiene* when hydrologic conclusions are to be drawn using RR modelling: the hydrologic
knowledge injected by the model developer must be transparent and must flow with minimal loss or corruption all the way
from the processes of model development and testing, into the application of the model, then into the production of simulations,
and finally into the conclusions drawn. One way to perform studies involving selective and real ignorance is simply to
document comprehensively what is assumed known, so that the ignorance can be deduced when necessary. For convenience,
such documents will be called *knowledge documents* and the method will be called *knowledge documentation*. To give a
practical example of selective ignorance, the entire fields of philosophy and epistemology are rendered here into the following
single knowledge statement in everyday English: *It is necessary that the work is common sense, transparent, repeatable by
others and testable*. Knowledge documentation is the method used here to maintain hygiene. Panel B in Fig. 1 shows the
scope for the knowledge document created later: it includes everything in Panel A in Fig. 1 up to and including "performance"
(NSE in the figure is the Nash Sutcliffe Efficiency; Nash and Sutcliffe, 1970). Essentially, the flow of hydrologic knowledge
is made visible and tangible by being rendered in the form of statements in everyday English.




The *hydrologic information content* of an RR record is large and diverse. Experience of studying RR records show that one storm hydrograph can look like another, and one catchment can respond in much the same way as another. There would, in fact, be little point in RR simulation modelling if such *similarity in time and place* could not be detected in RR records. Here, two days will be said to have hydrologic similarity if they have similar forcing (and, if relevant, also have similar runoff).


A hydrologist might defend conclusion 3 in Sect. 1 using an argument along the lines that snowmelt is sometimes visible in the RR records as the presence of sustained runoff in the absence of precipitation. For want of a better word, such a fingerprint in an RR record, whether it is in visual form or just as a set of data, will be called a SITH (*Something Interesting To Hydrologists*). Clearly, then, a storm runoff hydrograph can be a SITH, as can the timeseries of antecedent rainfall for a given

day. The experiment run by Crochemore et al. (2014) is probably a good starting point for anyone interested in what hydrologists see in RR records: a large number of hydrologists were asked to look at a set of simulated hydrographs and assess their quality.

The day of the year is a trivial SITH which can be extracted from the dates in RR records. It is the basis for the Peasant's

Model (Garrick et al., 1978): the runoff on a simulation day is set to the mean for the runoff observed on the same day of the year in other years. This is a *parameterless model*. It can be described as a *time–matching RR model* because the simulation day is matched with a set of other days, and the runoff observed on these other days is used in calculating the simulated runoff. If the SITHs used in a time–matching RR model are sophisticated enough so that their (dis)similarity can be measured, then potential matching days can be ranked by similarity and only the best matches used when calculating the simulated runoff.

Such sophisticated time–matching RR models are essentially *nearest neighbour models* (see Karlsson and Yakowitz, 1987). See also see the *ghost RR modelling* in Ewen and O'Donnell (2012) where the "ghosts" visible in simulations are echoes of previous storms.

When using only the minimum data set, *performance measurement* is the only mechanism by which hydrologic conclusions

can be drawn: *performance measurement is the gatekeeper to hydrologic conclusions*. In such work, performance measurement simply means measuring how well and/or reliably a simulation reproduces some SITHs of interest. The mean square simulation error for an entire simulation is the classic example of a *general performance measurement* (i.e. a single, full duration, measurement for which each SITH is an entire runoff hydrograph). Note that the NSE is linear in the mean square simulation error. Traditional lumped RR models such as GR4J (Perrin et al., 2003) are based on man–centuries of

accumulated community effort in the art and practice of calibration against general performance measurements, so are well suited to give a *predictive benchmark* (Shmueli, 2010) in the form of an upper limit for the NSE achievable for a given catchment.





## 3. Method

The details of the method are as follows. Building everything from scratch, and using knowledge documentation, a hydrologic
modelling study for UK catchments is undertaken in which simulations are produced and analysed. The study is an integral
part of a knowledge experiment in which statements of hydrologic knowledge lead to a set of parameterless time–matching
RR models. The NSE is used as a general measure for performance and the RR modelling uses a custom–designed MH which
is a layman who takes an interest in the weather and river flows and therefore knows some basic hydrology (e.g. a river
fisherman). One of the aims in the analysis is to measure the importance of the hydrologic knowledge held by the MH. This
importance is measured in terms of its impact on the NSE.

## 4 Theory

It would be all too easy to create a knowledge document which is simply a gateway to the tip of an enormous untestable cone–
shaped bulk of arguments and citations. Therefore: (1) the statements in knowledge documents must be designed to be taken
at face value; and (2) the hydrologic conclusions drawn from the documents must depend on taking the statements at face
value. Note that knowledge documents can be inspected by specialists from other fields, and will remain useful over time
because they can be re–examined by hydrologists and others in the future, when more is known. The authors of such documents
have nowhere to hide. An extreme form of knowledge documentation is used here in which new knowledge is added as a
statement in everyday English and this is then translated into other such statements, and these translated into other such
statements, and so on until the final such statements define the required algorithm.


If each day in an RR record has its own SITH, the degree of dissimilarity between two days can be calculated as the difference
between their SITHs. Each such difference is a performance metric and fits the wide, general definition given by McMillan
(2021) for a *hydrologic signature*. The whole process of detecting similarity can be generalised in obvious fashions. For
example, each day can have many SITHs and similarity can be calculated as a function of differences. Note that the case for
claiming that the hydrologic information content in RR records is large and diverse rests on the fact that thousands of SITHs
can be generated for a record (e.g. several per day), and these can relate to many different things (e.g. evaporation, storms,
flow volumes, snow melt, errors, similarity itself, etc.).

### 4.1 Sensitivity

When working with only the minimum data set, a study of the flow of hydrologic knowledge will usually degenerate into a
study of sensitivity. Sensitivity in RR modelling has a few different guises. The relevant guise here is as follows. If a model
developer introduces constants which must be fixed precisely (e.g. fixed at 74.2, say, rather than at an imprecise equivalent
such as 100), it seems reasonable to assume that the need for precision has an explanation. Clearly, then, if a model developer
must fix a constant precisely, the MH should know why (i.e. the MH should have the appropriate hydrologic knowledge). In





the experiment run here, the risk that any hydrologic knowledge is misused or missing is minimised by eschewing data fitting and checking if the model developer must fix any constant precisely. This makes it possible to draw conclusions which relate to hydrological knowledge. It is important to note, though, that any conclusions drawn from such an experiment, no matter how that experiment is designed or performed, will have a narrow context. This is a consequence of performance measurement being the gatekeeper to conclusions: any conclusions drawn will implicitly depend on how performance is measured. Strictly, then, it is correct to say that the conclusions drawn from the experiment run here relate to hydrological knowledge in the

context of general performance measurement using the NSE.

Sensitivity can play a complicated and misleading role in exercises involving data fitting. The main point to note about J&H's experiment is that by the act of measuring relative parameter errors the experiment was transformed from being a simple exercise in data fitting into an experiment on complexity in the context of general performance measurement using the mean

square simulation error (or the NSE). Looking dispassionately at the experiment, the MH in the modelling can be characterised as a statistician dilettante–hydrologist (albeit one created in a seminal work of scientific hydrology). This characterisation is based on noting that data fitting was used and that the main concept involved relies on little more than noting that runoff recessions look like exponential decay.

## 4.2 Time–Matching RR Modelling


Given that the simulations here are for 26 years and include 7 leap days, the maximum possible size of a list of matches for any given simulation day is 9,497 days (26×365+7). If this full list was used, the resulting time–matching RR model would give an NSE of zero because it would be identical to the benchmark model implicit in the definition of the NSE (i.e. it simulates the runoff as being constant at the mean rate observed over the simulation period). This model would be affected by what can

be called self–contamination, because the runoff simulated for day $d$ depends (slightly) on the runoff observed for day $d$. To avoid all risk of self–contamination, a 729 day self–contamination exclusion window will be used here which extends from 364 days before the simulation day to 364 days after, inclusive. The name *Trivial RR Model* is given to the model which uses the full list in conjunction with this exclusion window.

One of the time–matching models used here is the *Seasonal RR Model* which results when the following new statement is added to the Trivial RR Model: "Runoff is seasonal". The full set of statements which entail are given later, but it is instructive to work through the process without worrying about generating statements. Various translations are possible for the new statement. For example, a translation might introduce the concept of metrological seasons, leading to statements to the effect that each day is associated with only one of four time classes. The actual translation used introduces the concept of a seasonal–

scale (i.e. 91 day) inclusion window which is centred on the relevant day of the year. In a simulation run for $N$ years using the Seasonal RR Model, the simulated runoff for each day would be the mean for the runoff observed on a total of *91(N-2)* other





days. Note that this total includes the effects of the self–contamination exclusion window (that is where the factor *N-2* comes from) but neglects any end effects which arise for simulation days which lie near the start or end of the simulation period.

Once all the exclusion and inclusion windows have been applied, if the days remaining in the list can be ranked in terms of the quality of the match then the best *10* days can be picked and only these used when calculating the simulated runoff. Note that there is insensitivity to this sample size (i.e. *10*) so there is very little risk that fixing the sample size at 10 is associated with missing hydrologic knowledge. Tests show that, in the experiment, any sample size between 3 and 100 would be fine. The inbuilt mechanism for this insensitivity is that the quality of the matches tends to fall off only slowly with the sample size (which helps explain why the sample size could be as high as 100), whereas the incremental gain in accuracy from using a

mean falls off very rapidly with the sample size (which helps explain why the sample size could be a low as 3).

**5 Data**

Daily records were abstracted for 38 UK catchments (Table 1) for the period October 1978 to December 2005, inclusive, using publicly available data from the UK National River Flow Archive (NRFA; National River Flow Archive, 2019) and the UK

Met Office (Met Office, 2017). The catchments range in size from 12.4 km$^2$ to 1480 km$^2$. Some are at the coast, some inland, and some in the uplands, and there is a wide range of different types of response to rainfall. To create the necessary data sets for RR, catchment boundary and daily flow data were combined with daily gridded rainfall data.

**Table 1 The 38 catchments.**

| Symbol | Code | Name | Sq. km |
|--------|------|------|--------|
| A | 27035 | Aire at Kildwick Bridge | 282.3 |
| B | 55013 | Arrow at Titley Mill | 126.4 |
| C | 24004 | Bedburn Beck at Bedburn | 74.9 |
| D | 36003 | Box at Polstead | 53.9 |
| E | 53017 | Boyd at Bitton | 47.9 |
| F | 52010 | Brue at Lovington | 135.2 |
| G | 31010 | Chater at Fosters Bridge | 68.9 |
| H | 42008 | Cheriton Stream at Sewards Bridge | 75.1 |
| I | 37005 | Colne at Lexden | 238.2 |
| J | 39020 | Coln at Bibury | 106.7 |
| K | 22001 | Coquet at Morwick | 569.8 |
| L | 67018 | Dee at New Inn | 53.9 |
| M | 28046 | Dove at Izaak Walton | 83.0 |





| Symbol | Code | Name | Sq. km |
|---|---|---|---|
| N | 27042 | Dove at Kirkby Mills | 59.2 |
| O | 28008 | Dove at Rocester Weir | 399.0 |
| P | 39028 | Dun at Hungerford | 101.3 |
| Q | 48003 | Fal at Tregony | 87.0 |
| R | 26003 | Foston Beck at Foston Mill | 57.2 |
| S | 25006 | Greta at Rutherford Bridge | 86.1 |
| T | 31025 | Gwash South Arm at Manton | 24.5 |
| U | 60006 | Gwili at Glangwili | 129.5 |
| V | 41022 | Lod at Halfway Bridge | 52.0 |
| W | 29003 | Lud at Louth | 55.2 |
| X | 55014 | Lugg at Byton | 203.3 |
| Y | 28031 | Manifold at Ilam | 148.5 |
| Z | 38003 | Mimram at Panshanger Park | 133.9 |
| a | 43006 | Nadder at Wilton | 220.6 |
| b | 32006 | Nene/Kislingbury at Upton | 223.0 |
| c | 45005 | Otter at Dotton | 202.5 |
| d | 54016 | Roden at Rodington | 259.0 |
| e | 23006 | South Tyne at Featherstone | 321.9 |
| f | 33029 | Stringside at Whitebridge | 98.8 |
| g | 44006 | Sydling Water at Sydling St Nicholas | 12.4 |
| h | 50001 | Taw at Umberleigh | 826.2 |
| i | 54029 | Teme at Knightsford Bridge | 1480.0 |
| j | 60010 | Tywi at Nantgaredig | 1090.0 |
| k | 27034 | Ure at Kilgram Bridge | 510.2 |
| l | 53009 | Wellow Brook at Wellow | 72.6 |

## 6 Experimental design


The knowledge document developed here is for the entire first part of the knowledge experiment and not just for the RR modelling used in the experiment. The document comprises Tables 2–7. Given that the document is designed to be taken at face value it will not be discussed in any detail in the main text. KERR (the Knowledge Experiment RR Model) lies at the centre of the flow diagram in Panel B in Fig. 1. The locations of the tables relative to the model are shown in the diagram.

Knowledge flows from the MH and Developer into, within, and out of the model. Given that there is an experiment, there is



an Experimenter. The Experimenter designs and controls the experiment, so their knowledge flows through the entire experiment. Note that when obtaining results for drawing conclusions, KERR is sometimes replaced by one of its three simplified versions: the Trivial, Seasonal and Wetness Models.

## 6.1 Experimenter's Knowledge

The knowledge document must stand on its own feet. The Experimenter's knowledge (Table 2) therefore includes the essential elements from the discussion earlier. If success is to be claimed for the simulations, then requirements c and d in statement EK6 must be met. If success is to be claimed for the experiment as a whole, requirements a and b in EK6 must also be met. These relate to knowledge.

**Table 2 Experimenter's Knowledge (EK)**

| Label | Statement |
| --- | --- |
| EK1 | It is necessary that the work is common sense, transparent, repeatable by others and testable. |
| EK2 | This work uses the concept of selective ignorance, so only knowledge which appears in tables such as these is assumed known. The tables must contain only statements in everyday English. |
| EK3 | Parameterless modelling is to be used, so although statements can include text constants such as "365" they cannot include text parameters such as "leakage factor b" which would require evaluation for each catchment. |
| EK4 | The RR modelling is to be consistent with the hydrologic knowledge held by a Modelled Hydrologist who is a layman who takes some interest in the weather and river flows (e.g. a river fisherman). |
| EK5 | The performance measure of interest to the Model User is the NSE for entire simulations. |
| EK6 | For success: (a) hydrologic knowledge must not be lost or corrupted; (b) none of the hydrologic knowledge statements can be redundant; (c) the predictive headroom, as measured against an independent traditional lumped RR model, must be small; and (d) each catchment must have at least one proxy catchment such that there is little loss in performance when a simulation is not based on the catchment's own runoff record but on the runoff record for the proxy. |

## 6.2 The MH's Knowledge

In terms of physical concepts, a river fisherman would likely know quite a lot about seasonality and wetness. The MHK statements in Table 3 are written in a form more compact than would usually be uttered by a river fisherman, but that does not

alter their meaning. Note, also, that the statements are complete in themselves: they imply nothing about, say, the nature or magnitude of things such as evaporation.

**Table 3 MH Knowledge (MHK)**

| Label | Statement | Source |
| --- | --- | --- |
| MHK1 | (Seasonal statement) Runoff is seasonal. | Layman |
| MHK2 | (Wetness statement) Runoff rate depends on the temporal pattern of antecedent rainfall. | Layman |





## 6.3 The Model Developer's Knowledge

The knowledge held by the Model Developer comprises common knowledge and knowledge about time–matching and RR records (Table 4).

**Table 4 Developer's Knowledge (DK)**

| Label | Statement |
|-------|-----------|
| DK1 | (Unpredictability statement) Runoff records are always unpredictable to some degree because they are an imperfect reflection of reality and RR modelling is an approximation. |
| DK2 | In the algorithms, each simulation day is matched to a set of other days. The simulated runoff is the mean for the runoff observed on those other days. If self–contamination is to be avoided, all the days in the set must lie at least 365 days from the simulation day. If the quality of match is known, the set contains only the 10 best days. |
| DK3 | The simplest familiar generic measure for the distance between a pair of series is the root mean square difference. |
| DK4 | The simplest possible measure for how wet it is over a period of days is the mean rainfall rate for those days. |
| DK5 | A season lasts around 91 days. |

## 6.4 Translations

All the translations are simple and transparent (Table 5; and see the demonstration in Fig. 2). Note that: (1) statement TK2 is a selectively very ignorant and simple translation for the phrase "temporal pattern of antecedent rainfall" which appears in MHK2; and (2) the patterns for rainfall and wetness associated with any day $d$ starts on day $d$ and run back to the first day of its self–contamination exclusion window (i.e. day $d-364$). It can be quite confusing looking at plots in which time runs backwards, so in Fig. 2 the patterns are plotted backwards so that time runs forward. This figure shows there is a powerful inbuilt mechanism for insensitivity to the pattern length: storms on or just prior to day $d$ have high wetness, whereas earlier storms have low wetness. As a result of this insensitivity, there is very little risk of there being missing hydrologic knowledge associated with fixing the pattern length at 365 days (as a rough check, the entire experiment was repeated successfully with the pattern length fixed at 100 rather than 365).






**Table 5 Translation Knowledge (TK)**

| Label | Statement |
|---|---|
| TK1 | (Seasonal) A day should not be matched with any day lying more than 45 days from itself in terms of day of the year (day 1 follows day 365 or 366). |
| TK2 | (Wetness) For any day, the "temporal pattern of antecedent rainfall" has the following elements: the rainfall for the day; the average rainfall for the period covering the day and the previous day; the average rainfall for the period covering the day and the previous two days; and so on until the final period is 365 days. |
| TK3 | (Similarity) The root mean square difference between the 365–element–long patterns of rainfall for any two days will be called their *rainfall pattern difference*. It is a measure for the difference in wetness between them, so is a measure for how dissimilar they are. |

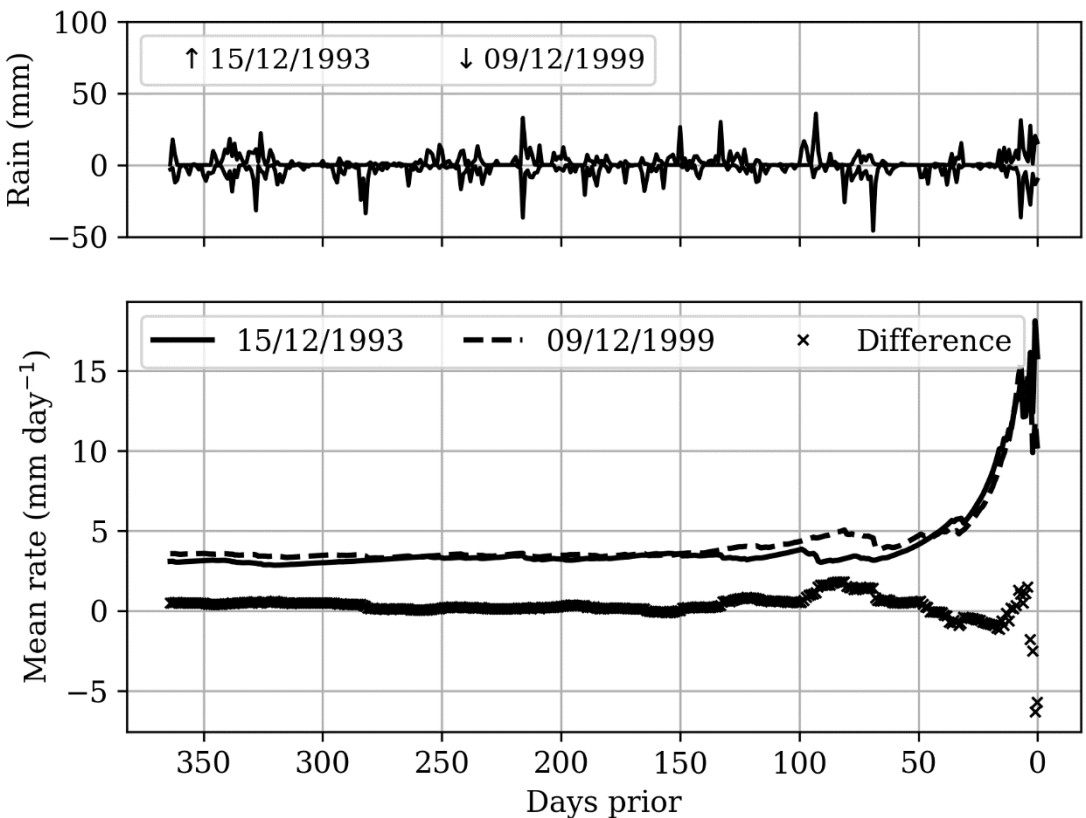

**Figure 2** The top panel shows the daily rainfall in the Aire catchment for the 365 day periods ending on 15/12/93 and 9/12/99. The bottom panel shows the antecedent wetness associated with the two days (calculated as mean rainfall rates, using the method described in statement TK2). It also shows the difference in wetness. According to TK3, the measure for the dissimilarity between the two days is called their *rainfall pattern difference*. It is the root mean square for the plotted differences (0.75 mm day$^{-1}$).



### 6.4.1 Similarity in Time

Although it is a slight detour, it is instructive here to use the translations above to explore similarity in time using a SITH
which comprises the wetness pattern for day $d$ and the runoff observed on day $d$. It will be assumed that two days are
hydrologically similar if their rainfall pattern difference is small and the difference between their observed rainfall is also small.
Figure 3 shows the most hydrologically similar days for the Aire catchment (i.e. catchment A). For every day, the 10 days
with the lowest rainfall pattern difference were found and then the best of these picked (i.e. the one closest in observed runoff,
irrespective of its rank in the set of 10). The two days in Fig. 2 are the most hydrologically similar for each other. Their
rainfall pattern difference is 0.75 mm day$^{-1}$ and their runoffs differ by only 0.2 mm day$^{-1}$. Their rainfall pattern difference is
the 8$^{th}$ smallest for 15/12/93 and 10$^{th}$ smallest for 9/12/99.

In Fig. 3, the blank space running diagonally from bottom left to top right is the effect of the self–contamination exclusion
window. There is some horizontal and vertical alignment of blank spaces, especially around 1996. This is associated with
prolonged drought. The effect of seasonality can be seen in the figure in the form of alignments of dots (note that these
alignments arose naturally: seasonality statement MHK1 was not used in finding the similar days). The four anomalous
horizontal alignments of dots associated with 1996 arise because the drought–affected days match best to days lying within
four narrow time periods. This figure gives some insight into the nature and (large) extent of the hydrologic information
content in RR records.



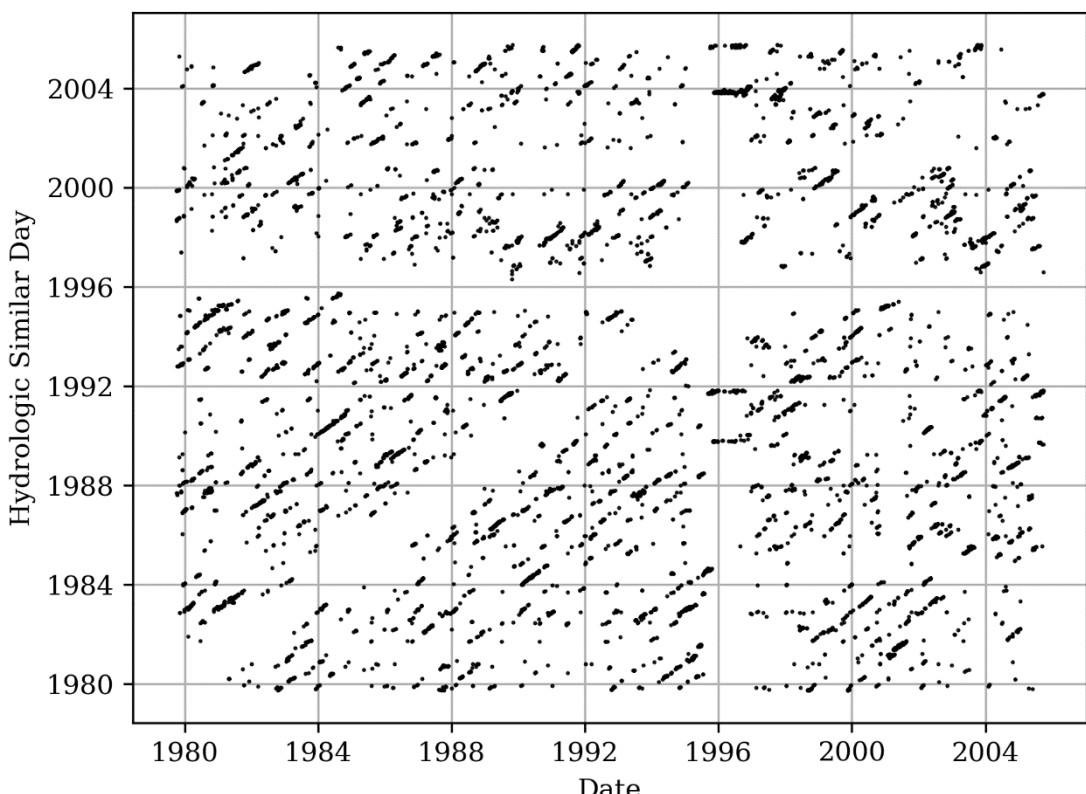


**Figure 3 Hydrologic similarity in time: best matching day for the Aire catchment.**

## 6.5 Algorithms

The statements for the algorithms are given in Table 6. It should be possible to reproduce all the knowledge experiment
modelling using the tables. Note that a few runoff values are missing from the records, so steps were taken to skip over these
(the rainfall data are complete). The Model User's knowledge is given in Table 7.





**Table 6 Algorithms**

| Label | Statement |
|---|---|
| A1 | (Trivial Model) Using DK2, accept every possible day apart from those lying in the self–contamination exclusion window. |
| A2 | (Seasonal Model) As A1, but also exclude the days which lie more than 45 days–of–the–year from the simulation day. |
| A3 | (Wetness Model) As A1, but then select the 10 days with the smallest rainfall pattern difference. |
| A4 | (KERR Model) As A2, but then select the 10 days with the smallest rainfall pattern difference. |

Fortran 2008 was used to build the models. The following code in the language R was written when testing the modelling. It is optimised for simplicity and clarity, not speed, and is for calculating the simulated runoff (sim) for day $d$ in a KERR simulation. The observed rainfall is in $p$, the observed runoff in $q$, and $md$ is the list of matchable days (i.e. the days remaining after accounting for all the exclusion and inclusion windows). In the first line, the 10 best matches are found by sorting the matchable days by their rainfall pattern difference (RPD) and then taking the first 10 days in the sorted list. In the line starting

with "return", wetness is calculated as cumulative sums of rainfall, running backwards in time, and then the root mean square differences in wetness is calculated. The other lines are simply a device which exploits the capabilities of R to work with lists. This device gives compact code and avoids the need to loop over the matchable days.

```
sim[d] <- mean(q[md[sort(RPD(d,md),index.return=TRUE)$ix[1:10]]])
dummy <- function(a,b) {
    return(sqrt(mean(((cumsum(p[a:(a-364)])-cumsum(p[b:(b-364)]))/(1:365))^2))) }
RPD <- Vectorize(dummy,vectorize.args="b")
```

**Table 7 User's Knowledge**

| Label | Statement |
|---|---|
| UK1 | High NSE is required. |

**7 Results**

Typical results from a simulation run using the KERR Model is shown in Fig. 4. The NSE achieved for the 38 catchments for the simulation period 1st October 1979 to 30th September 2005, inclusive, are plotted in Fig. 5. For the experiment, the predictive benchmark for a catchment is the average NSE calculated for that catchment using the results presented in the



evaluation columns in the supplementary material supplied by Harrigan et al. (2018) for the GR4J Model (Perrin et al., 2003). In Fig. 5, the catchments are ranked according to this benchmark. Note that Harrigan et al. used data for the potential evapotranspiration in addition to the minimum data set, and did not use the NSE when calibrating the model parameters. This is not ideal, but these are the best independent results found (Deckers et al., 2010, was also considered). When working with the NSE, a change of 0.1 corresponds to a 10 percentage point change in the explained variance of the observed runoff. Given

the nature of this work, 0.1 seems reasonable as a resolution to adopt when drawing conclusions.

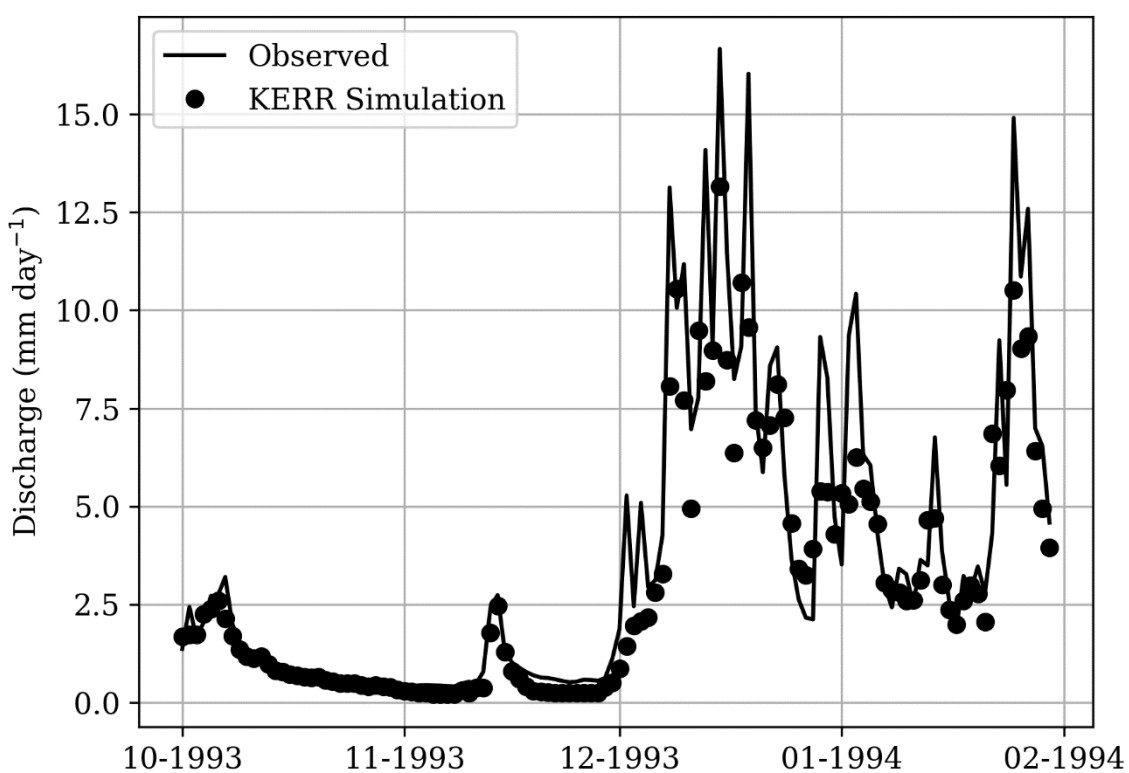

**Figure 4  Short section from the hydrograph for the Aire catchment (catchment A), demonstrating that the KERR modelling captures low flows as well as storm responses. The day of the highest peak (15/12/93) was discussed earlier for Figs. 2 and 3.**




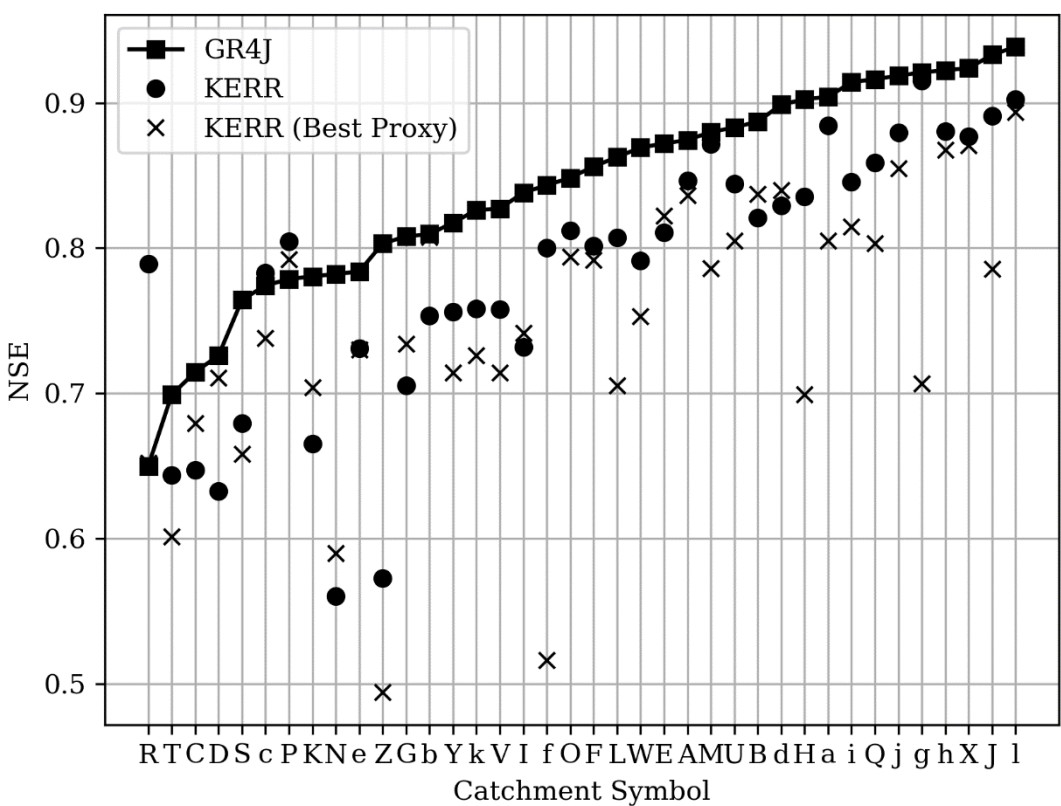

**Figure 5. NSE for GR4J and the KERR Model for the 38 catchments.**

**7.1 Transposability in time**

Table 8 gives the median NSE for the 38 catchments for all the models. For any catchment, the predictive headroom for transposability in time is the GR4J NSE minus the KERR NSE. The predictive headroom is 0.2 or less for 36 out of the 38 catchments, 0.1 or less for 33 out of the 38 catchments, and the performance exceeded that of the predictive benchmark for 3 out of the 38 catchments. This seems adequate.

For the layman's knowledge, on rounding to one significant figure, Table 9 shows that the relative importance for the seasonality and wetness statements are 1 and 6, respectively. This seems adequate as evidence for lack of redundancy. It may be that the statements are dependent: wetness is defined based solely on rainfall, and rainfall might vary seasonally for some catchments. The level of dependence could be examined by using each model in turn as the reference, but that would probably





need some theory (beyond the scope here) to allow meaning to be attached to differences in NSE when the differences do not
have a common datum.

**Table 8 NSE for transposability in time**

| Model | Median | Algorithm |
|---|---|---|
| Trivial | -0.005 | A1 |
| Seasonal | 0.194 | A2 |
| Wetness | 0.700 | A3 |
| KERR | 0.803 | A4 |
| GR4J | 0.852 | - |

**Table 9 Importance measured relative to the KERR Model**

| Statement | Mean | St. Dev. | Definition |
|---|---|---|---|
| Seasonality | 0.117 | 0.119 | Loss in NSE if seasonality statement MHK1 is forgotten |
| Wetness | 0.581 | 0.068 | Loss in NSE if wetness statement MHK2 is forgotten |
| Unpredictability | 0.216 | 0.091 | Maximum gain achievable in NSE if the unpredictability statement DK1 is wrong |


### 7.2 Transposability in place

It is required that each catchment must have at least one proxy catchment such that there is little loss in performance when the
matching is to the runoff record for the proxy catchment rather than to the runoff record for the simulated catchment. A total
of 1406 (i.e. 38×37) proxy catchment simulation were run using KERR (the self–contamination exclusion window is not
needed so was not used), and the best proxy catchment found for each catchment. The NSE when using this best proxy
catchment is plotted in Fig. 5. The median NSE is 0.75. Note that there is no loss in performance for 10 out of the 38
catchments, and a loss greater than 0.2 for only 2 catchments. This seems adequate.



# 8 Summary

1. Knowledge hygiene in Rainfall–Runoff (RR) modelling requires that the hydrologic knowledge injected by the model
developer should be transparent and should flow with minimal loss or corruption all the way from the processes of model
development and testing, into the application of the model, then into the production of simulations, then into the conclusions
drawn, and finally into the decisions made based on the simulations and conclusions

2. The fundamental problem of understanding the flow of hydrologic knowledge in RR modelling is a conundrum of great
importance to both engineering hydrology and scientific hydrology.

3. In exploratory scientific work, simple concrete examples can serve as references, benchmarks, exemplars, a basis for
generating hypotheses, and as supports when developing a line of argument.

4. In the concrete example created here, the flow of hydrologic knowledge into runoff simulations and performance
measurement is made visible using a custom–designed method (knowledge documentation) applied to custom–designed
parameterless RR modelling which uses only the minimum data set: i.e. catchment–average rainfall, the runoff at the catchment
outfall, and the catchment area. It is made visible in the form of a set of statements in everyday English.

5. The hydrologic information content of an RR record comprises thousands of SITHs (Something Interesting To Hydrologists
in the form of a visual fingerprint and/or the data associated with the fingerprint). The SITH used in the parameterless
modelling applies to a day and is simply a wetness pattern derived from the antecedent rainfall pattern for the day. The
hydrologic signature which is the difference between two of these patterns (called here the rainfall pattern difference) is a
measure for the dissimilarity of the days.

6. If a RR model was a hydrologist (the Modelled Hydrologist, MH) it would know only a limited amount of hydrology. It
can be said that an MH is selectively ignorant of hydrology. The experiment run by Jakeman and Hornberger (1993) on the
complexity warranted in RR modelling was discussed, given that it is the closest published work to that presented here. Using
the terminology of Klemeŝ (1986a), the MH in that experiment is a statistician dilettante–hydrologist.

7. The MH created here is a layman who takes an interest in the weather and river flows (e.g. a river fisherman). Using this
MH, a median Nash Sutcliffe Efficiency (NSE) of 0.80 was achieved for daily modelling for 38 UK catchments, and 0.75
when using RR records for proxy catchments.





8. The importance of hydrologic knowledge was measured in terms of the loss in NSE when statements of hydrologic knowledge were "forgotten". The layman has only two pieces of hydrologic knowledge: runoff is seasonal, and runoff rate depends on the temporal pattern of antecedent rainfall (i.e. wetness). The relative importance for these are 1 and 6, respectively, for a set of 38 UK catchments.

9. The relative importance is 2 for hydrologists' knowledge about unpredictability (specifically, knowledge that RR records are always unpredictable to some degree because they are an imperfect reflection of reality and RR modelling is an approximation).

10. The three pieces of hydrologic knowledge given in 8 and 9 above are adequate if the aim is simply to achieve high values
of NSE. Note that this conclusion, and the conclusions above about importance, and those drawn by Jakeman and Hornberger, all apply only in the context of general performance measurement using the NSE.

11. There are three constants in the modelling: 365, 91 and 10. Where relevant, checks were made to ensure that the act of fixing these constants did not need (hidden) hydrologic knowledge.

**9 Discussion**

The detailed conclusions from this work are given in the summary in Sect 8. RR modelling lies at the very core of catchment engineering hydrology and catchment scientific hydrology, yet it remains a conundrum. Science itself is a conundrum and keeps shifting and evolving, so it is inevitable that RR modelling is, at some level, a conundrum. That, though, is not a reason or excuse for failing to have hydrologic laws at the catchment scale. Laws might emerge soon from big data and artificial
intelligence, but might not. A little light was shone here. This light is in the form of: (1) disruptive, unsettling discussions about the nature of RR modelling; and (2) the creation of a simple, complete, concrete example for how hydrologic knowledge held by a model developer ends up affecting simulation performance.

For decades there has been a kernel idea (the wetness kernel) in RR hydrology. Here, it appears as a statement from a layman:
"runoff rate depends on the temporal pattern of antecedent rainfall". It's more familiar forms are as a leaky reservoir, an antecedent precipitation index, and an (inverted) unit hydrograph. A traditional RR modeller who focusses on implementation details and mathematical structures would immediately spot the kernel in the modelling here and could with a little thought link it to each of the familiar forms. The (valuable) unusual feature here is that, compared to traditional implementations, the kernel has been deconstructed and de-parameterised. It is odd that the wetness kernel has not evolved into a physical
hydrologic law or been used to support a richer, perhaps non-physical, hydrologic law of some kind.

## 9.1 Future Work

There is huge scope for future work. It would, for example, be interesting to try and bridge some of the gap between an MH which is a layman and an MH which is acquainted with a panel of experienced hydrologists. Perhaps this would qualify as
what Beven (2001) described as "modelling as collective intelligence". One of the most surprising results here is that successful proxy catchment simulation does not require scaling, alteration or transformation of any kind. This indicates strong and fundamental similarity. If a search was mounted for a hydrologic law which corresponds to this similarity, the task would amount to discovering the hydrologic knowledge needed to pick a good proxy.

Section 6.4.1. gave a template for future work on hydrologic similarity. One interesting question about the direct use of similarity in RR modelling is whether it gives different simulations compared to traditional modelling even if both use the wetness kernel. Note that the hydrologic knowledge resulting in leaky reservoirs amounts to knowledge that runoff hydrographs can be decomposed efficiently into sets of exponential decays, but the layman's knowledge is used simply as a "release agent" which releases part of the hydrologic information content of the RR record.

## Code Availability

The essential part of the code was given in the main text as a snippet in the language R.

## Data Availability

The catchment boundary and daily river flow data can be obtained from https://nrfa.ceh.ac.uk/ and the gridded UKCP09 daily
rainfall data from https://catalogue.ceda.ac.uk/uuid/319b3f878c7d4cbfbdb356e19d8061d6.

## Author contribution

JE and GO discussed this work in detail over several years. JE performed the experiment. GO selected the catchments and tested the data and experiment. JE prepared the manuscript with contributions from GO.

## Competing Interests

The authors declare that they have no conflict of interest.





## Acknowledgements

This work was funded under the Natural Environment Research Council project "Susceptibility of catchments to INTense RAinfall and flooding" (SINaTRA; NE/K00896X/1) and the Engineering and Physical Sciences Research Council project "Urban Flood Resilience in an Uncertain Future" (EP/P004180/1). We are grateful to Elizabeth Lewis for preparing the data and to Enda O'Connell and Russell Adams for very helpful comments on various versions of this manuscript. Data from the UK National River Flow Archive and the UK Met Office were used.

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
