# Peer review of "If a Rainfall–Runoff Model was a Hydrologist"

_Hydrology and Earth System Sciences, 2021_

## Community Comment (CC5)

I will reiterate my concerns about this paper again, since they have not been addressed - even superficially - by the previous replies. However, I will not respond to further comments - there is nothing that can improve this paper without a complete re-write beginning with at least a cursory literature review of the subject that is being discussed and then a rethinking of the primary message in the context of a more rigorous attempt to derive concepts. Any further dialogue along the lines of the comments below is fruitless. This is exactly the type of article that the peer-review process is supposed to catch, although this one is so obviously flawed (i.e., does not cite any literature from the domains that it speaks about), that I am a little surprised that the editor requested a review.

In detailed response to responses by the authors (author's responses in purple):

CC3 says: "*You have chosen to focus on one (small) aspect of my comment and ignore the rest.*"  In fact, we focussed on a claim that our model has absolutely no lasting value whatsoever to anyone, and a claim to the effect that the subject of the title of the paper is a redundant conception.  These claims are central to two of the three general concerns you highlighted in your review.  The claims are not well founded (they lack insight) and some readers may read the review but not the paper.  We therefore responded to the claims as soon as we could.

As explained in my original comments, the point of the comment about the skill of your model is that your proposed philosophy and writing suggestions lack *empirical* (in addition to theoretical and philosophical) support. As an empirical demonstration of the proposed philosophy and technical writing suggestions, the empirical results *degraded* relative models developed using current standard practice.

A sense of proportion and fairness is needed in discussing the third highlighted concern (philosophy).

You are writing on an academic subject (model realism) that is centuries old and one of the most widely studied problems in academic philosophy, and perhaps in academic history (e.g., Quine, Hemple, Cartwright, among hundreds or thousands of others). It is ok to have opinions about this subject, but unless those opinions contribute to the academic discussion in a rigorous way, they do not belong in peer-reviewed journals.

In CC4, in the name of philosophy, you try and shout down (cancel) hydrologists who use intuition creatively in hydrology.

The point of peer review is to keep non-academic papers from being published, and this current paper represents the epitome of the need for peer review. You are not being "cancelled" - you simply have not done what is necessary to participate in the academic discussion that you are trying to write about. You do not have a right to publish your meandering, uninformed, non-academic opinions in a peer review journal -- this is what blog posts are for, and having a paper rejected because you failed to

follow basic protocol for writing an academic paper (e.g., citing even a single source from the primary topic you are attempting to publish on) is not an injustice.

Cannot intuition, and the insight it brings, not simply be appreciated and be adapted for use for the general good.

Intuition is not a substitute for rigorous epistemology, which is one of the oldest and most mature fields of study. Intuition is not a substitute for all of the components of an academic article that are missing from this paper: (i) a relevant literature review, (ii) formal logical or theoretical foundations, (iii) logical derivation of the proposed philosophy, (iv) supporting empirical results, etc. Intuition is sometimes useful, but it is not a sufficient basis for formal academic contributions, especially on a subject that is as mature and well-studied as this one.

It never crossed our minds that a reader or reviewer would persist in the notion that we are somehow trying to reinvent technical writing or are engaged with what you describe as "*changing how we write scientific papers*".  Neither did it cross our minds that a reader or reviewer would persist in assuming we propose the use of everyday English other than in scientific exploration, and then only when it is useful and practical (our background is physically-based, distributed RR modelling, where the documentation runs to hundreds of pages of text, equations and diagrams).

Your (only) actionable suggestion in the paper is to use knowledge tables written in "everyday english" (this is a direct quote from your paper). I have a hard time imagining that you failed to anticipate that people would read the exact words you wrote in the paper.

Additionally, this paper does not make a formal (either theoretical or empirical) case for moving away from the standard methods for technical writing that you alluded to here, which are extremely effective. Personally, I would rather have a 100-page model documentation that gives rigorous, reproducible methods and statements of theory and assumptions than tables with sentences in "plain english". However, I'm not going to argue with you about the validity of your technical writing suggestion because - frankly - neither of us are qualified to discuss the subject of technical writing or philosophy of communication at a level sufficient for peer-reviewed journal. And this is the entire point of my criticism - you substituted uneducated intuition for academic rigor, and this kind of writing does not belong in peer-review.

The paper gives a science-based solution to a real-world problem: benchmark links between hydrologic knowledge and performance are needed as a basis for measurements related to engineering decisions.

There is no science in this paper. No hypothesis was tested (in my original review, I attempted to - generously - treat the new model development as a hypothesis test of the new philosophy, but the reviewers did not even recognize that this is what I was doing in their replies).

There is irony in that a serious attempt to be clear about what is assumed known in reaching the solution is attacked on philosophical grounds, especially given L128-131. Also, any discussion of the solution, or how it was arrived at, must take into account that in L180-181 we explicitly allow for permanent review.

There have been many (thousands) of serious attempts at reconciling the model realism problem both in philosophy journals and domain science journals. The problem with this paper is precisely that it is *not* a serious attempt to do that - it does not even recognize, let alone build on any of the existing work on the subject. It is just the intuition-based musings of people who have not even made a cursory attempt to do a literature review on the topic they are attempting to publish on.

We have been thinking about what might be covered if a discussion section is to be added to the paper. The predictions are for the numbers in runoff records, so in the context of the paper the records are reality. Say there are three regions in a space: physical reality (i.e. the river catchments), hydrologic knowledge and performance. The paper is about a single mapping from hydrologic knowledge to performance. Other mappings are not discussed, such as mappings to or from physical reality, or back from performance. One-to-many, many-to-one and many-to-many mappings are not discussed. To the extent that it can be helpful, such mappings could be described in a discussion section in terms of common philosophical concepts which interest RR modellers.

This would just be more non-rigorous, intuition-based musings that would not improve the paper.

The 2nd paragraph in CC4 is grossly unfair. It seems to be a reaction to this text from AC1: "*One of the points made in the blind validation work is that models and modellers must be seen as a package (Ewen and Parkin, 1996). Our experience is that hydrologists running an RR model sometimes forget the nature of the model. Sometimes it is treated as a statistical black box. The worst case is when the model is treated as if it is reality, and it is implicitly assumed that there are no constraints on what can be concluded from the resulting simulations.*" The term "black box" seems to have been lifted from this text and its meaning adjusted to fit your case. The reality is that RR models are often run as a general resource, well outside the control of model developers (you seemed to have assumed that the text is about model developers running their own models). Some models run as a general resource have considerable complexity, and this can lead to belief in simulated detail (including spatial variations in response) or in all the available energy being spent on the sheer effort of parameter calibration against one or a few statistics (i.e. treating the model as if it is a black box).

I disagree that there are model users out there who do not understand that models are approximations and have limited domains of applicability. But again, I will not engage in this discussion beyond stating this disagreement because it is irrelevant to the question of whether this article is publishable. My dis/agreements with the authors on matters that are not quantifiable and/or not derivable (e.g., how humans interpret or how they apply models) should not (and does not) factor into my review of this paper. The paper fails (abjectly) due to lack of academic rigor.

---

## Author Comment (AC3)

The reviewer's comments are in italics and the responses in regular text.

*It is always interesting to read one of John's papers, and this one is no exception.*

Thank you.

*Given the current penchant for throwing data into the black boxes of machine/deep learning algorithms and declaring success without producing much in the way of understanding, the more thoughtful approach presented here, in a highly original way, is of value.*

The work with such "black boxes" is interesting.

*I did find the paper not easy to read in places, with some sweeping generalisations and somewhat limited reference to previous work (see below). And the end result is limited to a really simple non-parametric modelling approach that appears to only reflect the basic minimum of hydrological knowledge, that flows reflect today's rainfall and some index of antecedent conditions (here represented as only by matching the pattern of average rainfalls over the last n days, in a least squares sense without any allowance for autocorrelation in those values).*

*It sort of works (even better than a calibrated rainfall-runoff model for some of the catchments and mostly not much worse in terms of NSE). It sort of works for transfer from a proxy catchment (but see comment below on this). But there are perhaps some issues of hydrological knowledge that are somewhat glossed over.*

(A) When compared against a model like GR4J, KERR does "sort of work". That is a little surprising given it is parameterless and is based on trivial hydrologic knowledge. In fact, it works well enough to be seen as a reasonable RR model, and the experiment essentially shows that a model of a layman can give a reasonable RR model. If KERR was a model of us (i.e. the authors), rather than a layman, its median NSE might be higher as a result of our understanding of many of the technical details mentioned in the review. It is important to note, though, that the aim is not to do the best RR modelling possible, but the best knowledge experiment possible, based on a model of a layman. That might help explain why the work seems to be based on sweeping generalisations and limited references to previous work.

*There will be uncertainties in the data. These are mentioned but then ignored. But can be significant – e.g. where event runoff coefficients are highly variable and go greater than 1 (see Beven and Smith, 2015; Beven, 2019).*

*This implies that it might be useful to allow for uncertainty in the mean prediction – you can after all pattern match to get an ensemble of possible values which could be treated as a first estimate of a pdf reflecting uncertainty.*

For convenience, the responses to all the questions about uncertainty are given here.

(B). With hindsight, a brief note on uncertainty should have been included in the Future Work section of the paper (Sect 9.1). The RR records used are a snapshot taken at some time before 2014, when the work started. The records change over time. There is now the CamelsGB set, where a different rainfall product was used and for which at least one of the 38 catchments has a revised

rating curve. Relative to CamelsGB, the data errors in the original set are therefore known and their impact on the knowledge experiment could be calculated. There is a sampling problem in that only 38 catchments were used and a fixed period of time selected. Such things could be addressed in future work.

(C). The terminology used in the paper is that there is "predictive headroom" in that GR4J gives a higher NSE, suggesting there is scope for finding better hydrologic knowledge because there seems to be some unexploited structure within the RR data. "Unpredictability" includes what comes under predictive headroom, plus data errors, plus errors in rating curves, etc. The measure for unpredictability is simply the mean deficit w.r.t. perfection, so is the mean difference between 1 and the NSE values achieved using KERR. That mean is what is in the third row in Table 9, and that row is where the value 2 comes from which is attributed to unpredictability in L18 in the abstract and L425 in the summary. Text to this effect will be added to the results section.

(D) In L236 the number of best days is fixed at 10. In the context of the work there is no uncertainty about this. The value 10 was picked and was found to be entirely successful given that testing shows that there is insensitivity to picking 10. It is possible to calibrate the number of best days or to construct numerical schemes that use ensembles. In the context of the knowledge experiment, though, nothing meaningful would be gained by doing that. The case is the same for fixing the pattern length at 365 days.

*The use of daily data is convenient in terms of getting hold of the data but will be subject to discretisation issues in small catchments (where the peak occurs in the dayaffects the volume for that day) and autocorrelation issues in larger catchments (every day is here treated as independent, even on recessions).*

We all have to work with the data we can get hold of. In other work, experiments using KERR with sub-daily data worked fine.

*Snow is mentioned, but then neglected. It may be relatively unimportant in most UK catchments perhaps, but one of the outcomes from the Iorgulescu and Beven (2004) attempt at a similar non-parametric data-based predictor based on CART also with different rainfall period inputs, was that the classification identified anomalous periods associated with delayed snowmelt. That this might happen is hydrological knowledge is easily stated in English!*

An experiment could be run for a layman who takes an interest in snow. Iorgulescu and Beven (2004) will be cited.

*For the transferability in space, a brute force approach to finding the best KERR model is taken but checking all the catchments in the data set and picking the best as the donor site. That cannot be used if a catchment was treated as ungauged (when the proxy basin transfer is actually required) and really does not seem to be making too much use of hydrological knowledge (though the difficulty of transferring response characteristics using catchment characteristics or model parameters is, of course, well known). But our model hydrologist could perhaps be expected to know that there is an expectation that catchments of different scales might involve different processes, or catchments in hard rock wet areas of the west might be expected to be different to chalk catchments in the south east. So, in respect if the title of the paper the words very naïve might need to be added before*

*hydrologist (and indeed the MH is referred to elsewhere as a layman or angler rather than someone with better hydrological knowledge).*

The fact that a layman can be modelled gives some hope that less naïve hydrologists can also be modelled. The proxy catchment results show there is a deep-seated similarity at play: given any catchment A, a catchment B can be found such that when the catchments have the same rainfall pattern they have the same specific discharge. This deep-seated similarity might help with the ungauged catchment problem.

*The paper does not mention that we are often wanting to simulate the potential effects of future change. If that change is only to the inputs then the proposed strategy might work, perhaps with some degradation if the processes change. If, however, if it is change due to reforestation or NFM measures or other changes, then it could be used as a baseline to compare with future observations, but not as a simulator of a changed future (and indeed the changes might be within the uncertainty of the predictions if that was assessed in some way).*

If there is a there is deep-seated similarity, it could help with the usual extrapolation problem (e.g. the proxy catchment might have some periods with higher rainfall intensity than the target catchment). The work gets quite far considering it is for a layman. Change impacts for reforestation etc. are outside the scope of the work. Perhaps, though, there is some potential in adapting the similarity in time method used to create Fig. 3 to the study of trends or anomalous variations in the matching of days.

*Which then, of course, raises the question of what might happen if the MH had access to that committee of experienced hydrologists (or even inexperienced hydrologists – see the tale of the hydrological monkeys in the Prophecy paper cited). That experience might lead them to think more in terms of model parameters than direct use of data (Norman Crawford, Sten Bergstrom, and Dave Dawdy are examples from quite different modelling strategies but all were known for their skill in estimating parameters for models, including for ungauged sites .... though there may have been some potential for positive bias in tweaking and reporting results there). And there are instances of committees of experienced hydrologists not doing that well in setting up models and even getting worse results as more data were made available (see the Rae Mackay et al. groundwater example from NIREX days).*

The committee would need a chairman, and the work would stand or fall on how the chairman is selected. There is no reason why parameters cannot be used, but a method would be needed to track the consequences through to the conclusion drawn. The concrete example in the paper drives a spike in the ground and gives a place to measure from, for the combination of NSE and hydrologic knowledge. Note that the value of the example comes from the fact that is for a combination, and not simply for the NSE on its own. There is no reason why performance should improve when more is known or is assumed known, unless a monotonic rising relationship is built in by design. Jakeman and Hornberger built such a relationship into their performance measurement (i.e. the performance for the modelling as a whole and not simply for hydrographs).

*I would suggest that the authors could make more of the difficulties of going further with more experience and knowledge about catchment characteristics. It is an argument for their KERR*

*approach – but I would also suggest that the KERR approach also be extended to reflect the uncertainty to be expected as a result of that simplicity.*

The Future Work section (Sect. 9.1) will be extended to suggest how the method could be taken further and to discuss uncertainty.

*Some specific comments*

*L37 Best available theory – but there is also the issue of whether that theory is good enough when it differs from the perceptual model of the processes.*

Yes.

*L56. There seems to be a lot of overlap between what is referred to here as hydrological knowledge and the concept of a qualitative perceptual model. Both need to be simplified to make quantitative predictions (and often do so in ways that conflict with the perceptual model because of what is called here selective ignorance).*

A mention of the overlap will be added to the text. No limit is placed on the nature of hydrologic knowledge. It could, for example, be quantitative or be about how performance must be measured.

*P121. There were earlier suggestions of this approach, e.g. Buytaert and Beven, 2009, or even the donor catchment approach of the FSR/FEH.*

Agreed.

*L128. This is analogous to the Condition Tree concept in Beven et al, CIRIA Report C721 (also Beven and Alcock, 2012) that results in an audit trail to be evaluated by others.*

Noted.

*L132. Performance really ought to take account of uncertainty in the data (see earlier comment and papers cited in Beven, 2019)*

See responses B-D above.

*L137. But catchments that look very similar can also respond quite differently – even if mapped as the same soils/geology. We have an example from monitoring two small catchments on the Howgills. So issue of requisite knowledge here is when such small scale variability might integrate out (or not) – this was discussed in the 80s as a representative elementary are concept – eg. Wood et al.1988; also papers on when variability in stream chemistry starts to integrate out).*

The text at L137 just notes that RR modelling utilises similarity in time and place (without specifying the nature of that similarity). The nature of the similarity, in as far as it is apparent in RR records, is considered in other parts of the paper.

*L149. I think the "peasant's model" was suggested by Eamon Nash in modelling the Nile before this.*

We will look for this.

*L166. But again that upper limit will also definitely depend on the uncertainty and inconsistencies in the observations.*

See response C above.

*L182 nowhere to hide – exactly the point made for the Condition Tree / audit trail*

Noted.

*L336. Why not use an ensemble here to add in some uncertainty to the process?*

Have assumed that this is L236.  See response D above.

*L394. But it is only a match to rainfall pattern – is there no additional knowledge that could be used? In the case of expected greater autocorrelation in large catchment for example (or extension to shorter time steps in small catchments) perhaps the last few predictions of flow might be useful (avoiding just doing 1 step ahead forecasting, though such a model with updating does also represent the forecasting model to beat – e.g the Lambert ISO model, see RRM book)*

See response A above.

*P425 relative importance is 2???? Not clear.*

See response C above.

*P426. Always upredictable – see the inexact science paper again*

See response C above.

*P456. Does not require scaling – There is an expectation that processes change with increasing scale, and that specific discharge becomes less variable with increasing scale, and generally less in moving from headwaters to large scales, so does this imply that this is compensated by a decline in mean catchment rainfalls so that the power on the area scaling is low, or simply that the variability is within the uncertainty of the predictions so does not have too great an effect on NSE? There are past studies on scaling with area that might be treated as hydrological knowledge here.*

Perhaps this similarity in place boils down to a simple thing related to the evolution of the geometry of flow paths, rather than directly to some form of compensation between scales and processes and dynamics.

Wood, E.F., Sivapalan, M., Beven, K.J. and Band, L. (1988), Effects of spatial variability and scale with implications to hydrologic modelling. J. Hydrology, 102, 29-47.

Buytaert, W and Beven, K J, 2009, Regionalisation as a learning process, Water Resour. Res., 45, W11419, doi:10.1029/2008WR007359.

Beven, K. J. and Alcock, R., 2012, Modelling everything everywhere: a new approach to decision making for water management under uncertainty, Freshwater Biology, 56, 124-132, doi:10.1111/j.1365-2427.2011.02592.x

Beven, K. J., and Smith, P. J., 2015, Concepts of Information Content and Likelihood in Parameter Calibration for Hydrological Simulation Models, ASCE J. Hydrol. Eng., 20(1), p.A4014010, doi: 10.1061/(ASCE)HE.1943-5584.0000991.

Beven, K. J., 2019, Towards a methodology for testing models as hypotheses in the inexact sciences, Proceedings Royal Society A, 475 (2224), doi: 10.1098/rspa.2018.0862

Iorgulescu, I and Beven, K J, 2004, Non-parametric direct mapping of rainfall-runoff relationships: an alternative approach to data analysis and modelling, Water Resources Research, 40 (8), W08403, 10.1029/2004WR003094

---

## Author Comment (AC4)

The reviewer's comments are in italics and the responses in regular text.

*In the manuscript "If a Rainfall-Runoff Model was a Hydrologist" by Ewen and O'Donnell, a set of parameterless rainfall-runoff models are developed in an experiment which aims to quantify the importance of the knowledge contained by the model itself. To make this knowledge explicit, the rainfall-runoff model is personified as a layman with an interest in the weather and river flows. The model is parameterless and relies on time-matching based on the similarity of the simulation day with a set of other days from the historical data record. The model performance of the developed KERR model is overall just slightly lower than the GR4J model performance for a set of UK catchments. A main finding of the paper is the strong relative importance of the temporal pattern of antecedent rainfall. This concrete model development example supports a broader and more philosophical discussion on hydrologic knowledge and laws within rainfall-runoff models.*

*The manuscript addresses relevant scientific questions on the role of hydrologic knowledge contained in rainfall-runoff models on model performance. However, at first read, I found the manuscript to be confusing in how it is structured and in its balance between the broader philosophical discussion and the very concrete, simple and specific modeling experiment. Which aspects of the specific example should or can we apply in more complex traditional rainfall-runoff modeling, is it the knowledge documentation through personification of the rainfall-runoff model?*

*I hope that the comments below will help improving the manuscript.*

Thank you. The comments will be helpful in improving the manuscript.

(A) We accept that readers find the paper confusing on first reading. Some linking and summarising sentences will be added to brings out the scope and structure of the work and paper. The paper is unusual in that its scope is huge: broader philosophical questions are discussed, many definitions created, methods created, RR models developed, a knowledge experiment run (which is essentially a test for the hypothesis that **An RR model can be a model of a hydrologist**), and a concrete example created which gives a benchmark which can be used as a basis for generating hypotheses.

(B) The scope for what can be meant by "more complex traditional rainfall-runoff modelling" is huge. It extends from basic scientific studies, through model development and testing, to model use in a variety of ways and for an endless list of purposes. It is up to modellers to pick out what they find useful in the paper, given their own circumstances and requirements.

*General comments:*

*1) The aim of the paper is not clearly stated in the abstract, I would suggest to explicitly add it. The aims are described in L70 and later L110 and in L174, however, throughout the manuscript it remains unclear what is exactly meant by 'corruption' of hydrologic knowledge flows within RR modelling. Could you clarify this further?*

There is discussion of the scope and structure of the work in response para. A above. The aim is to create the concrete example (i.e. an RR model which is a model of a hydrologist), as indicated in the title and stated in L14-15 in the abstract and in L83. That aim needed a lot of support in the form of definitions and methods and discussion of broader philosophical questions.

(C) The general point about corruption is that the simulations and conclusions from RR modelling sometimes do not depend wholly or exactly on what the model developer knew or assumed when

creating the model. There are three parts to how a conclusion is reached: (1) the design of the model by the model developer; (2) how the model actually works; and (3) the model user drawing conclusions from simulations. Clearly, there can be corruption in part 2: e.g. missteps, approximations, and in translations when an algorithm is created from knowledge. The classic example of approximation is when numbers representing physical properties lose their meaning, such as when spatial differential equations for runoff are solved on a coarse grid. There can also be problems with parts 1 and 3, including problems which arise when model users create knowledge flows out of thin air. For example, KERR does not know about the physical reality of river catchments: the RR records lie at an outer boundary (see Fig.1) and are essentially a set of numbers. If in part 3 KERR is used to draw concussions relating explicitly in some way to realism then it will involve the misallocation of simulation responses to knowledge relating to realism (i.e. the model user creates a knowledge flow out of thin air). Parameter calibration is a confounding factor and can play a role in the general set of problems relating to the misallocation of simulation responses to hydrologic knowledge, and the misallocation of hydrologic knowledge to simulation responses. If and where that is seen as corrupting the flow of knowledge would depend on exactly what is done during calibration and exactly what conclusions are drawn from the results from calibration.

*2) The manuscript does not contain a dedicated Conclusion section. Concluding remarks are provided in the Summary of Section 8. However, I think it would help the reader to include a dedicated conclusion section which specifically links back to the aims of the study. This would help to clarify the main message/focus of the manuscript.*

See response para. A above.

*3) The experiment was performed for a set of UK catchments. Could you discuss the application of the developed parameterless models and the conclusions drawn on the importance of wetness in the light of different climatic zones?*

It would not make sense to study the importance of wetness using modelling that gives poor performance as a result of neglecting something that is very important in a region, such as snow or water resources management or drought.

*4) How could the proposed methodology of quantifying the importance of hydrologic knowledge held by the MH on model performance be applied in more traditional rainfall-runoff modeling?*

The method was created as a means to an end, for the knowledge experiment. This question therefore goes well beyond the scope of the paper. Given the nature of more traditional rainfall-runoff models, one way forward might be to use algorithmic differentiation to estimate the point sensitivities of performance to parameters (or the sensitivity of conclusions to parameters, if the conclusions are numerical or can be forced to be numerical).

*5) Hydrological modeling is often used in practice to quantify the effect of change in a catchment (e.g. land use). In science, hydrological modeling is often used to increase our understanding of catchment functioning. Both would be difficult using the proposed approach of the parameterless model, could you please elaborate on this?*

The parameterless modelling was created as a means to an end, for the knowledge experiment. This question therefore goes well beyond the scope of the paper. Off the shelf lumped RR models are of limited use if the aim is to understand catchment functioning. The best way forward is probably to work with black box models (e.g. deep learning RR models) and look for general laws or specific behaviours, but to do this taking into account the richness and value of what is already known. One conclusion which can be drawn from the paper is that any attempt to use lumped RR modelling to increase the understanding of catchment functioning must take very great care over the selection and use of performance measurement. Note that when measured in terms of the median NSE, UK catchments function exactly as laymen think they function.

*6) The manuscript includes several references to the study of Jakeman and Hornberger (1993). However, a short summary of the main aspects of this paper in relation to the current paper seems to be lacking.*

Agreed.

*7) The way the work is presented is sometimes confusing. For example, in Section 4.2, the Trivial and Seasonal RR models are presented. Later in Section 6, an additional Wetness model is mentioned. In Table 6, also the KERR model is presented. Perhaps, it would be good to clarify this in Section 4.2 and in the Method section 3 so that the reader has a better understanding of the main approach.*

See response para. A above.

*Specific comments*

*L5: Could you add here why personification can also be instructive?*

Yes.

*L10: Simplification of complex systems is inherent to modeling, but I guess you want to quantify how and which of the knowledge contained in the model mostly affects model performance?*

Yes.

*L11: What do you mean by classic MH?*

The classic structure for conceptual RR models comprises little more than a few leaky reservoirs: this structure is in widespread use and has been so for a few decades. This structure brings with it a need for parameter calibration. The combination of a few leaky reservoirs and a need for parameter calibration is the classic form for conceptual RR modelling.

*L17: I found the sentence with "the relative importance is measured as 1 and 6" rather confusing. Do you mean: antecedent wetness is 6 times more important than seasons in simulating runoff in a time-matching modeling approach? I would suggest rephrasing this sentence (also in the Summary section).*

You understood the meaning correctly.  We will rephrase the sentence.

*Figure 1: Although mentioned in the caption, it was at first read not entirely clear to me that the numbers refer to the knowledge statements of the different Tables, I would suggest rephrasing the caption to clarify.*

Agreed.

*L66-67: Could you elaborate further on this?*

One interpretation of what Jakeman and Hornberger did was to create a compound performance measurement which included the NSE and a measure for the degree of overparameterization.  They then manually adjusted the model structure to find good structures which maximise the compound measure.  It is the fact that they used an appropriate compound measure which makes their conclusions about complexity useful.

*L70: As mentioned before, what do you exactly mean by 'lost or corrupted' hydrologic knowledge flows within RR modeling?*

See response para. C above.

*L72: What do you exactly mean by science is an "activity and attitude"?*

Baker (2017) should have been cited.  Also see the response below (L73) .

*L73: Could you elaborate further what you mean by the significant deficits, difficulties or dangers?*

It can be quite daunting when an engineering decision has to be made based on scientific understanding.  The best first step is often simply to force forward to get a concrete solution of some sort, even if it is for a simplified problem, and then work from there (it gives perspective).  Getting to a solution involves paying attention to any deficits (especially perceived deficits in understanding), finding solutions to difficulties (e.g. not having a model to work with) and being aware of what might have gone wrong or might go wrong.

*L85: It is not clear to me what you mean by "not well behaved", could you clarify?*

It is not a simple, orderly flow.  There can be loops and jumps.

*L111: aim (3) is not entirely clear, could you elaborate on "the need to draw valid hydrologic conclusions"?*

The reason why hydrologists use RR modelling is to reach some conclusion or other, whether it is simply about what the flow is or will be, or about some process or assumption, or the basis for some engineering decision.  The aim and hope is that the conclusion is valid.

*L134: In 'traditional' hydrological modeling, would you also recommend describing equations in the form of statements in everyday English? How does this relate with the more commonly provided model descriptions and equations?*

KERR was designed for use in a prototype machine which analyses text and numerical tables. That is why everyday English was used as the mechanism for keeping control of the knowledge (rather than using equations or unnatural English which simply mimics logic sequences and equations). The approach was described as "extreme" in L182. Less extreme methods, with a mix of text and equations, would probably work just as well if there is no need for analysis using any specific type of tool.

*L138: with "Here," do you mean: in the models being developed in this study?*

"Here" means in this paper.

*L141: could you specify conclusion 3 in Sect. 1, it is unclear to me to which point you are specifically referring to.*

It is L34.

*L174: when you mention "one of the aims", I would find it helpful to also have a recap of the other aims of the research.*

Agreed.

*L198: I am not sure to fully understand what you mean by "the MH should know why", could you clarify this part?*

Knowledge is needed if the value of a constant has to be set, and that knowledge will clearly be of considerable importance if the value has to be set precisely. If a model was a hydrologist then the hydrologist would be acutely aware of that knowledge.

*L201: "from such an experiment", do you mean an experiment without data fitting?*

Yes, and with fixed constants that did not need to be set precisely.

*L259: in contrast to the Trivial and Seasonal models, the Wetness model was not introduced earlier.*

See response para. A above.

*Fig2: Rain is a flux and should therefore have unit [L/T], I assume here it is mm/d. It is somewhat confusing to show negative values on the y-axis of the top panel. It would be clearer for the reader if rainfall pattern difference was also explained in the text describing Fig 2.*

Rain is the depth accumulated in a day. The -50 will be changed to 50. A pointer to the figure legend will be added to the text.

*In the paragraph 313-319, the horizontal alignments around 1996 are explained twice.*

One mention is for alignment of blank spaces and the other for alignment of dots.

*Table 6: is the KERR model a general name for the Trivial, Seasonal and Wetness models and a fourth model? Could you please clarify?*

KERR is not a general name. It is the name for the model containing algorithm A4 (see Table 6).

*L355: could you elaborate: "when drawing conclusions" on what?*

When drawing conclusions about the median NSE and impacts on the median NSE.

*Table 9: conclusion based on the third statement "Unpredictability" are not explained in the text of Section 7.1. This statement only comes back in the Summary of Section 8. Could you please elaborate on this finding already in Section 7.1?*

Text will be added to Sect. 7.1. If there was perfect predictability then the NSE for all catchments could be 1.0, and if this was the case the mean rise in NSE for the 38 catchments would be 0.216 (see final row in Table 9). The importance of unpredictability is therefore 2 when compared to the 6 for wetness and 1 for seasonality.

*L429: Here, I would suggest to explicitly mention "wetness, seasonality and unpredictability" to clarify: "the three pieces of hydrologic knowledge given in 8 and 9".*

Agreed.

*L449: Could you elaborate further on what you mean by the wetness kernel and discuss more in detail the related hydrologic law?*

The notion of wetness linked to accumulated rainfall is a core idea that appears in several different forms in RR modelling, going back decades. In theory, it could be treated along the lines of Darcy's law and a simple classification process linked to the parameters for, say, leaky reservoirs or accumulations of rainfall. Note that, in the first instance, the underlying problem can be reduced to a cluster analysis.

*Discussion: how important is personification of RR models? This seemed to be an important focus point at the start of the manuscript.*

The concrete example is a hypothesis test for the concept of the MH (see response para. A above), so personification is a focus all the way through the paper, not just at the start. Personification gives

context to the benchmark created for the combination of the NSE and hydrologic knowledge, and that benchmark is generally useful (it is a benchmark for a layman). The notion of the personification of RR models does not, in itself, give a direct solution to anything. However, it may have an effect on what RR modellers think, how they think, and what they do. Until RR modelling is on a far better footing (e.g. using hydrologic laws at the catchment scale), it is difficult to know what will ultimately be found to be important.

*Typos:*

*L14: a MH instead of an MH*

*L228: meteorological instead of metrological*